# Towards real-world monitoring scenarios: An improved point prediction method for crowd counting based on contrastive learning

Rundong Cao⬚*, Jiazhong Yu, Ziwei Liu, Qinghua Liang

China Tower Corporation Limited, Beijing, China

* caord@chinatowercom.cn

## Abstract

In open environments, complex and variable backgrounds and dense multi-scale targets are two key challenges for crowd counting. Due to the reliance on supervised learning with labeled data, current methods struggle to adapt to crowd detection in complex scenarios when training data is limited; Moreover, detection-based methods may lead to numerous missed detections when dealing with dense, small-scale target groups. This paper proposes a simple yet effective point-based contrastive learning method to alleviate these issues. Initially, we construct contrastive cropped samples and feed them into a convolutional neural network to predict head points of each image patch. Based on the classification and regression loss of these points, we incorporate an auxiliary supervision contrastive learning loss to enhance the model's ability to differentiate between foreground heads and the background. Additionally, a multi-scale feature fusion module is proposed to obtain high-quality feature maps for detecting targets of different scales. Comparative experimental results on public crowd counting datasets demonstrate that the proposed method achieves state-of-the-art performance.

## Introduction

Crowd counting is a crucial research topic in the fields of public safety and video surveillance, with broad application prospects. For instance, by analyzing crowd counts, we can estimate the aggregation state of crowds in large stadiums, squares, conference centers, and entertainment hubs. This enables situational awareness of crowd dynamics, thereby preventing incidents such as stampedes and mass brawls. Additionally, by examining crowd density distributions, we can assess the commercial value of specific locations or regions, facilitating the formulation of reasonable market planning and development strategies. In traffic management, counting the number of people at intersections, major thoroughfares, and public transportation hubs helps

**Data availability statement:** The UCF_CC_50 dataset analyzed during the current study is available in the https://www.crcv.ucf.edu/data/ucf-cc-50/. The ShanghaiTech Part A and Part B dataset analyzed during the current study is available in the https://github.com/desenzhou/ShanghaiTechDataset.

**Funding:** No funding was received to assist with the preparation of this manuscript.

**Competing interests:** The authors have no competing interests to declare that are relevant to the content of this article.

devise effective traffic management measures, such as security personnel allocation and evacuation plans.

Technically, crowd counting, often referred to as crowd density estimation, involves using computer vision techniques to estimate the number of individuals in a given image. Crowd density estimation based on computer vision has been studied for nearly 30 years. Early methods, primarily based on shallow visual computing techniques, have been quite comprehensive [1]. Representative methods include pixel feature statistics [2,3], texture analysis [4], head template features [5], background estimation and Expectation Maximization (EM) [6], Gaussian Process Regression (GPR) [7], support vector machine regression based on feature points [8], multi-camera information fusion [9], real-time head counting based on feature points [10], multi-output regression models based on feature mining [11], Bayesian regression [12], unsupervised Bayesian detection [13], semi-supervised elastic net [14], among others. These methods have significantly advanced crowd counting techniques. However, due to their reliance on manual feature extraction, their performance in adapting to complex scenarios remains suboptimal, falling short of practical technical standards.

With the widespread application of deep learning methods in computer vision, crowd counting techniques have seen substantial advancements [15,16]. Under the deep learning framework, existing crowd density estimation methods can be roughly divided into three categories: detection-based methods, density map-based methods, and regression-based methods. Detection-based methods treat pedestrians as visual targets, using deep learning techniques to obtain a series of target boxes, from which the total number of individuals in the image is counted. For instance, head visual feature learning based on cascade Adaboost and convolutional networks [17] and adaptive head candidate region generation based on scale maps [18] improve the accuracy of pedestrian detection by enhancing the network structure design for crowd scenes. Additionally, frameworks such as Faster-RCNN [19] and the YOLO series [20–26] are widely applied in crowd counting systems, enhancing pedestrian counting accuracy across various scenarios and crowd sizes. However, these methods heavily depend on the accuracy of detection boxes, leading to significant counting errors when pedestrians are densely packed and occlude each other.

Density map-based methods achieve crowd counting primarily through traditional kernel density estimation or density map estimation based on deep learning. Within the deep learning framework, methods such as dilated kernel convolutional networks [27], spatial divide-and-conquer networks [28], shallow feature dense attention networks [29], dilation rate adaptive convolution [30], and local counting maps [31] have improved the quality of crowd density map estimation. Some methods focus on generating high-quality crowd density maps, such as Fusion Count [32], adaptive density map generators [33], patch-level density map generation networks [34], and dynamic crowd density map refinement networks [35,36]. These methods have enriched the technical means of crowd counting. However, density map-based crowd counting methods in crowded areas are affected by cluttered backgrounds, crowd scale, perspective effects, target occlusion, and density loss [37–39]. Furthermore, density map methods struggle to directly obtain the location of each individual, impacting practical video surveillance applications.

Point-based methods use key points to locate heads or bodies, directly identifying individuals in the scene. Essentially, these methods belong to a localization approach. Song et al. [40] proposed a point-based joint crowd counting and individual localization framework, introducing density-normalized average precision and constructing a Point to Point Network (P2PNet) that directly predicts a set of point proposals representing head locations in the image. They employed the Hungarian algorithm to match predicted points with ground truth points, with matched points representing head locations, thus counting the total number of individuals. Additionally, methods such as point confidence prediction based on Transformers [41], self-attention-guided center point methods [42], bipartite matching point-supervised crowd counting methods [43], self-training methods based on point-level annotations [44], and Bayesian loss probability models [45] have improved the accuracy of pedestrian point prediction and localization. Despite providing precise individual locations, these methods depend on the network's feature extraction capabilities, often struggling with differentiating positive and negative samples and detecting heads of varying scales.

To address these issues, this paper introduces contrastive learning techniques into a head point prediction framework to enhance the model's ability to distinguish between positive and negative samples. During model training, a series of positive sample image patches containing heads and negative sample patches without heads are randomly cropped to construct positive and negative sample groups for contrastive learning loss calculation. This approach trains a network model that better distinguishes between head foreground and background. Additionally, considering the varying sizes of real crowd objects in video applications, a multi-scale feature fusion module is designed to enhance the model's feature extraction capabilities for head targets in real scenes. This method directly predicts head points in images during inference, thus counting the total number of individuals.

For current practical applications, since video surveillance requires crowd location information, we primarily consider the YOLOv7 target detection method [46] and the head point detection-based model method proposed in this paper. The comparison of detection box and point detection effects in dense crowd scenarios is shown in Fig 1. To improve the robustness of the model, more real-world training data and the use of data augmentation techniques have been

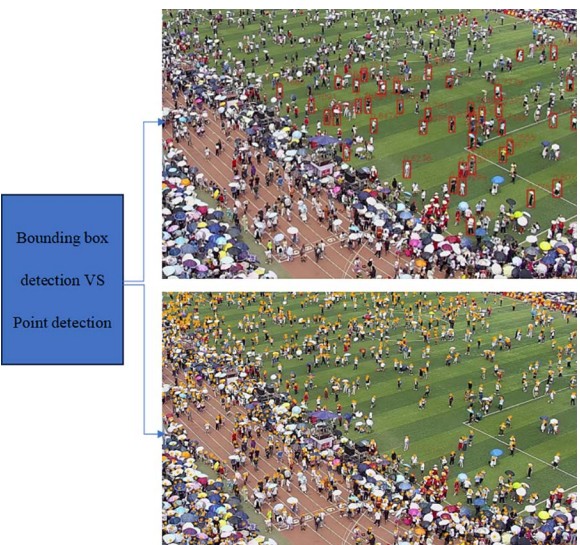

**Fig 1. The comparison between bounding box detection and point detection on the Tower dataset collected from practical applications.** The upper image shows the performance of the YOLOv7 bounding box detection method, while the lower image presents the results of our point-based detection method. As we can see, the point-based detection method performs better. All faces are blurred in Fig 1 for privacy preservation.

introduced. We could train the model with different lighting conditions, weather changes, and different viewpoints to make it perform well in real-world applications.

The main contributions of this paper include the following aspects.

- We propose a novel and effective method for crowd counting and localization, which achieves state-of-the-art performance on crowd counting datasets and has been widely applied in practical video surveillance scenarios.

- We propose a crowd feature representation network based on patch-supervised auxiliary contrastive learning, which fully leverages the local density characteristics of the crowd in scene images, enhancing the discriminative capability between head targets and background without adding extra inference burden.

- We introduce a multi-scale feature fusion module, designing a weighted cross-scale connection structure to aggregate features at different resolutions, thereby improving the model's ability to learn head features of varying scales.

## Related work

In this section, we review some recent works on crowd counting and contrastive learning. As detection-based methods and point-based methods can be summarized as localization-based methods, we discuss these two crowd counting approaches.

### Density map-based methods

For a given crowd image, density map-based methods aim to generate a density map and then sum the predicted density map to obtain the count [27–31,47,48]. Specifically, Li et al. [27] proposed the Congested Scene Recognition Network (CSRNet) model, which uses dilated kernels to provide a larger receptive field to generate high-quality density maps, thus achieving high-precision crowd counting. Liu et al. [47] introduced a nonlinear continuous counting quantization strategy and transformed the problem of sample block counting imbalance into a class imbalance of counting levels. Xiong et al. [28] proposed the Spatial Divide-and-Conquer Network (S-DCNet), whose core idea is to learn a counting classifier on a closed set and then extend it to open-set counting. Miaott et al. [29] proposed a Shallow Feature-based Dense Attention Network (SDANet), capturing multi-scale information through densely connected hierarchical feature maps, achieving crowd counting for static images. Bai et al. [30] proposed a dilation rate adaptive convolution operation, establishing a self-correcting supervision mechanism to improve the accuracy of crowd counting. Liu et al. [31] proposed a local counting map, constructing a scale-aware module, hybrid regression module, and adaptive soft region module to achieve high-precision counting regression by focusing on the difference between the global sum of the crowd count and the density map during testing. Liu et al. [48] transformed the counting problem into a sequential decision problem, using a weighing strategy from scales to achieve a crowd counting model based on deep reinforcement learning.

Most existing density map-based crowd counting methods mainly focus on crowd density estimation. Relatively few works focus on generating crowd density maps. Ma et al. [32] proposed a crowd counting model that extensively leverages representations learned during encoding to compute first-phase multiscale features, and its decoder further fuses these scale-aware features to generate the density map. Wan et al. [33] analyzed the impact of different density maps and constructed a density map refinement network. They built an adaptive density map generator, using annotation dot maps as input to learn the density map representation of the counter, generating ground-truth density maps. Xu et al. [49] extracted patch-level density maps through a density estimation model, introducing multi-polar center loss to automatically normalize each patch density map online, achieving density map clustering. Jiang et al. [50] constructed a multi-level convolutional neural network that adaptively learns multi-level density maps, each focusing on handling pedestrians of specific sizes and fusing them to predict the final output. Tian et al. [35] proposed a more intuitive and understandable Density Map Dynamic Refinement Network (DDRNet), consisting of a counter and refiner. The refiner, composed of convolution

layers with different dilation rates, iteratively refines and improves the quality of the density map using the counter's output as dynamic input. Liu et al. [36] constructed a dynamic fine density map network with a designed regional attention module (RAM) that adaptively adjusts the head size relationship in different positions of the dot map, refining existing ground-truth density maps through joint training of the counter and learnable refinement network.

These methods have advanced crowd counting techniques but have yet to meet practical requirements. Firstly, manually annotating crowd density maps is challenging. Additionally, these methods heavily rely on the quality of crowd density maps. In practice, the density maps generated by existing methods are easily affected by changes in head proportions due to the multi-scale nature of pedestrians in images, failing to reflect the actual size of heads in the image, thereby impacting counting accuracy.

## Localization-based methods

Localization-based methods utilize object detection techniques [51–53] for crowd counting, with the core idea of treating pedestrians or heads as visual targets and counting the crowd by locating these targets. Gao et al. [17] proposed using the cascade Adaboost algorithm to replace the candidate region generation module in the R-CNN framework [54], using convolutional networks to learn head visual features and employing a support vector classifier for pedestrian and non-pedestrian classification. Khan et al. [18] proposed using a scale map to generate scale-adaptive head candidate regions, followed by using convolutional neural networks for head detection. Rani et al. [19] first used Faster-RCNN [55] to detect heads and then built a pedestrian counting system. Additionally, the YOLO series object detection frameworks [20] have been widely applied in pedestrian detection and crowd counting tasks [21–26]. Sam et al. [34] developed a tailored detection framework for dense crowd counting by predicting head bounding boxes, using a top-down feature modulation strategy to better distinguish pedestrian targets and producing fine-grained predictions at multiple resolutions, reliably outputting head localization results from sparse to dense crowds.

Moreover, some point-based methods have been applied to crowd counting. The primary task of point prediction is to use supervised deep learning methods to achieve pedestrian target point regression based on pedestrian (particularly head) point annotations, thereby providing individual localization positions. Song et al. [40] proposed a Point to Point Network (P2PNet), a classic architecture among such methods, directly predicting a set of point proposals to locate head positions. Yuan et al. [41] constructed a Localization Guided Transformer (LGT) framework, a point-based model using regression heads and classification heads to simultaneously predict head point proposals and point confidence, providing more discriminative representations for high-quality density map estimation. Ma et al. [42] proposed a self-attention guidance-based crowd localization and counting network (SA-CLCN), using original point annotations from crowd datasets as supervision to train the network, predicting each head's center point coordinates and the crowd count. Liu et al. [43] proposed a bipartite matching-based point-supervised crowd counting method, matching annotated pixel points through bipartite matching to reduce the impact of incorrect point matching on counting performance. Wang et al. [44] proposed a novel self-training method, using point-level annotations and crowd-aware loss to guide network training, predicting pedestrian center points and sizes in crowded scenes. Ma et al. [45] proposed a density contribution probability model based on Bayesian loss and point annotations, where the training loss constrains not the value of each pixel in the density map but the expected count of each annotated point, improving crowd counting accuracy in dense scenes.

## Contrastive learning

The concept of contrastive learning was proposed as early as 2006 to learn invariant representations of patterns [56]. Subsequently, Khosla et al. [57] developed supervised contrastive learning, whose core idea is to cluster the embeddings of positive samples while separating the embeddings of negative samples. Supervised contrastive learning has significantly enhanced the learning ability of image feature representations. Recently, contrastive learning methods

have gradually been applied to pedestrian detection and crowd counting. For instance, Lin et al. [58] proposed an example-guided contrastive learning framework to guide feature learning. Under the contrastive learning framework, they used pedestrian appearance as a prior knowledge example dictionary, constructing effective contrastive training pairs and using the constructed example dictionary to evaluate the quality of candidate pedestrians. Chen et al. [59] proposed a discriminative feature learning framework for crowd counting, consisting of a Masked Feature Prediction Module (MPM) and a Contrastive Learning Module (CLM). MPM randomly masks feature vectors in the feature map, enhancing the model's pedestrian localization ability in high-density areas through a supervised reconstruction strategy. CLM brings the representations of pedestrian targets closer and pushes the background features away from pedestrian representations, increasing their discriminability. However, these methods do not solve the problem of crowd counting in both sparse and dense crowd scenarios. Unlike previous works, our method combines patch-level foreground and background contrastive learning with a point detection framework, providing more accurate results in various applications.

## Problem description

For images from wide-area surveillance scenarios that may contain crowds, let $N$ represent the number of people, and we use $p_i = (x_i, y_i)$, $i \in \{1, \ldots N\}$ to denote the center point of the $i$-th head. The set of real crowd points in the image can be represented as $P = \{p_i | i \in \{1, \ldots N\}\}$. The predicted set of head points by the model is denoted as $\hat{P} = \{\hat{p}_j | j \in \{1, \ldots M\}\}$, and the confidence score set of each head point is $\hat{C} = \{\hat{c}_j | j \in \{1, \ldots M\}\}$. Thus, the crowd counting problem can be described as ensuring that the predicted point $\hat{p}_j$ with high confidence $\hat{c}_j$ is as close as possible to the real point $p_i$, while the predicted number of people $M$ is as close as possible to the real number $N$. Therefore, our method design needs to simultaneously consider accurate category prediction and precise position regression. Given the varying head sizes in real scenarios and the presence of negative samples with features similar to heads, our model needs to have multi-scale feature learning capabilities and strong positive-negative sample discrimination.

## Our method

In this section, we propose a point-based crowd counting method incorporating contrastive learning, which consists of five key components: (1) Random cropping to obtain contrastive samples; (2) Backbone for extracting image features; (3) Multi-Scale Feature Fusion Module (MSFM) to improve the ability to detect heads of different sizes; (4) Multi-branch head modules for classification, regression, and projection; (5) Point matching during the training process and point prediction during the inference process.

### Contrastive learning network design

The objective of this section is to construct positive and negative samples for contrastive learning. Given a crowd image, we randomly crop a fixed number of 128 × 128 image patches, which may or may not contain people. Our key insight lies in aggregating the feature representations of crowd regions while separating those of background regions. By batching the image samples, we can highly likely obtain a series of samples that simultaneously include both crowd and background regions. These samples can then be utilized as positive and negative samples to establish a contrastive learning framework. For instance, when the batch size is set to 16 and the number of patches is 8, each training iteration will generate 128 samples, either containing or excluding human elements, which are applicable for the contrastive learning process.

Similar to existing crowd counting models [40], we can use VGG-16_bn [60] as the backbone network for extracting image features. As shown in Fig 2, by outputting three different levels of feature maps, we design a Multi-Scale Feature Fusion Module (MSFM) to obtain better feature representations of crowds of varying scales and distributions. Subsequently, based on the same output of the MSFM, we construct three different head branches. During the training phase, we adopt a linear projection layer as the projection head, and utilize three stacked convolutional layers interwoven with

ReLU activation as the classification head and regression head. During inference, only the classification and regression branches are maintained for point prediction.

## Multi-scale feature fusion module

The designed Multi-Scale Feature Fusion Module (MSFM) is shown in Fig 3. It takes three feature maps as input and outputs a fused feature map. The input feature maps $L_{in}$, $M_{in}$, and $H_{in}$ have sizes H/4xW/4, H/8xW/8 and H/16xW/16 respectively. The output feature map has a size of H/8xW/8. We first utilize FPN [61] to introduce top-down and bottom-up paths, obtaining three new feature maps with stronger semantics and finer information, namely L, M, and H. Considering the limitations of FPN in multi-scale feature fusion, we further introduce two additional connections. The mathematical description of the new feature maps is as follows:

$$H = Conv2d\left(H_{in}\right)$$

$$M = Swish(Conv2d\left(M_{in}\right) + Upsample(H)) \quad (1)$$

$$L = Swish(Conv2d\left(L_{in}\right) + Upsample(M))$$

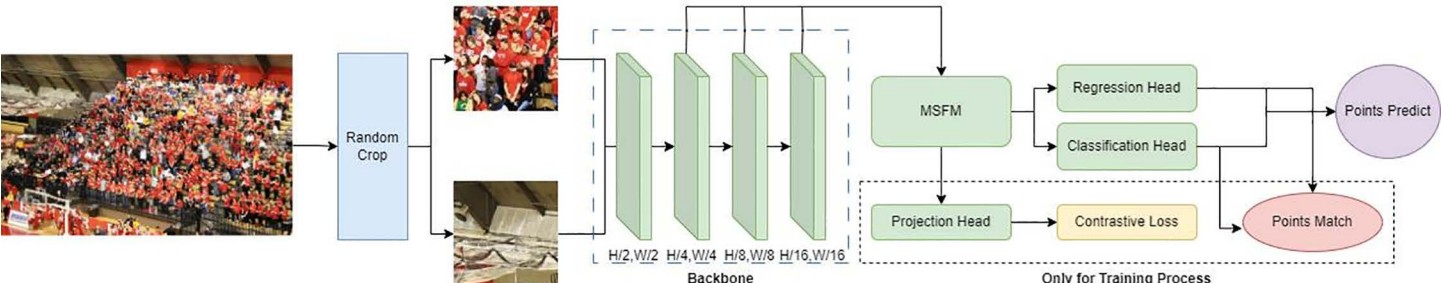

**Fig 2. Overall framework of point-based crowd counting with contrastive learning.** The blue dashed box includes four different-sized feature layers from the VGG backbone network. The backbone network section can be replaced with other structures such as ResNet. The dashed box containing the projection head, contrastive loss, and point matching is used only during the training process.

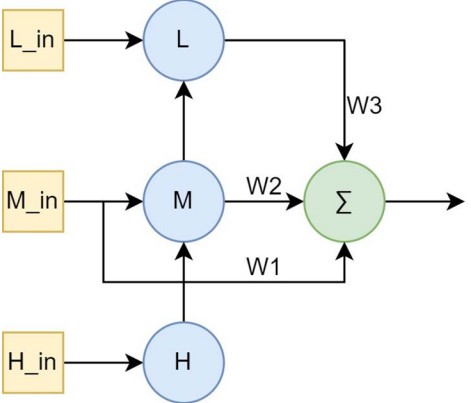

**Fig 3. Structure of the multi-scale feature fusion module.** L, M, and H represent Low-level, Medium-level, and High-level features, respectively. W1, W2, and W3 are learnable weights for features at different levels. The symbol Σ denotes the element-wise weighted summation operation.

where Conv2d is the standard two-dimensional convolution operation, Upsample is the upsampling operation, and the Swish activation function [62] is chosen for better performance, specifically defined as:

$$Swish(x) = x \cdot \frac{1}{1 + e^{-\beta x}} \tag{2}$$

Here, $\beta$ is a constant or trainable parameter, set a $\beta = 1$ in our network module.

For cross-scale connections, we additionally add a bottom-up path from the L level to the M level, using a MaxPool operation with stride 2 to adjust the output size of the L level to the same shape as the M level. Moreover, we add an extra edge connection in $M_{in}$ to sum operations of the same level as the M output, enabling the fusion of more features without incurring significant costs. Due to different input features at different resolutions, their contributions to the output features are typically unequal [63]. To address this, we perform a weighted summation operation on the inputs $M_{in}$, M, and L, with the final fusion formula as follows.

$$Output = Swish\left(W_1 * M_{in} + W_2 * M + W_3 * MaxPool(L)\right) \tag{3}$$

## Point matching and prediction

Similar to P2PNet [40], we use a one-to-one matching strategy for point matching. A set of fixed reference points $R = \{R_k | k \in \{1, \ldots K\}\}$ is introduced. These reference points are densely and uniformly arranged over the patch obtained from the input image crop, with coordinates represented as $R_k = (x_k, y_k)$. The classification branch is designed to output the confidence scores of these point coordinates, while the regression branch generates the offsets for these point coordinates. For a predicted point $\hat{p}_j = (\hat{x}_j, \hat{y}_j)$, the offset is represented as $(\Delta_{jx}^k, \Delta_{jy}^k)$, and the predicted point coordinates are calculated as follows.

$$\hat{x}_j = x_k + \gamma \Delta_{jx}^k$$

$$\hat{y}_j = y_k + \gamma \Delta_{jy}^k \tag{4}$$

where $\gamma$ is a regularization parameter used to adjust the magnitude of the offset.

For each point proposal in the set $\hat{P}$, a one-to-one matching strategy is employed, assigning them to the ground truth target set P. Based on the Euclidean distance and confidence score of these points, a pairwise cost matrix D is constructed, defined as follows.

$$D\left(P, \hat{P}\right) = (\tau \left\|p_i - \hat{p}_j\right\|_2 - \hat{c}_j) \tag{5}$$

where $\tau$ is a weighting parameter, and $\hat{c}_j$ is the confidence score of $\hat{p}_j$.

Using the above pairwise cost matrix, the Hungarian algorithm [64,65] is employed for matching. In our implementation, the total number of reference points exceeds the number of pixels in the image, ensuring that the number of predicted points exceeds the number of ground truth points. Thus, matched predicted points can be labeled as positive samples, while unmatched predicted points are labeled as negative samples.

## Loss function design

In our framework, there are three loss objectives: classification loss, regression loss, and contrastive loss. The first two are based on the matched and unmatched points obtained above, while the last one is calculated based on the deep feature maps $F_s$ output by the MSFM.

For the classification loss, we adopt focal loss [66] for dense target detection, which is described as follows.

$$p_t = \begin{cases} p & \text{if } y = 1 \\ 1 - p & \text{otherwise} \end{cases}$$

$$FL(p_t) = -\alpha(1 - p_t)^\gamma log(p_t) \tag{6}$$

$$L_c = \frac{1}{M}\sum_M FL(p_t)$$

where $p$ is the model-estimated probability for the label being 1, $\alpha$ is a weighting parameter to address class imbalance, $\gamma$ is a focusing parameter that reduces the loss contribution from easy examples and extends the range of examples receiving low loss. MMM represents the mini-batch size.

For the regression loss, we introduce MSE loss to supervise the point regression, defined as follows.

$$L_R = \frac{1}{N}\sum_{i=1}^{N}||pos_i - pos_{M(i)}||_2^2 \tag{7}$$

where $pos_i$ denotes the position of the ground truth point, $M(i)$ denotes the point matched with the ground truth point $i$, and N is the number of matched points.

The contrastive loss is calculated on the feature maps output by the MSFM. Since we use random cropping to obtain patches during training and input them into the network, the first dimension of the output feature map is the product of batch size and the number of patches, denoted as $m$. Let $m$ be the index of any sample in $m$, $A(m)$ be the index set excluding $m$ itself, and $P(m)$ be the index set of other aligned visual features with the same label as $z_m$. The contrastive loss for a training process is defined as.

$$L_A = \sum_{m \in M}\frac{-1}{|P(m)|}\sum_{p \in P(m)} \log \frac{exp(s_{m,p}/\tau)}{\sum_{a \in A(m)} exp(s_{m,a}/\tau)} \tag{8}$$

where $\tau$ is the temperature hyperparameter, and denotes dot product. $s_{m,p} = \frac{z_m^T z_p}{||z_m||||z_p||}$ is the cosine similarity between sample $m$ and sample $p$, $z_m$ and $z_p$ are the high-dimensional feature vectors of samples $m$ and $p$, respectively. Therefore, the overall loss function can be formulated as follows.

$$L_{tot} = \lambda_1 L_A + \lambda_2 L_C + \lambda_3 L_R \tag{9}$$

where the weights for these loss functions $\lambda_1$, $\lambda_2$, $\lambda_3$ are hyperparameters. In our experiments, we set $\lambda_1$ to 0.01, $\lambda_2$ to 1, and $\lambda_3$ to 2e-4.

## Experiment

### Experimental details

We utilize multiple publicly available crowd counting datasets, such as UCF_CC_50 [67], ShanghaiTech Part A and Part B [68], to demonstrate the superiority of our method. The UCF_CC_50 dataset contains 50 images with the number of people ranging from 94 to 4543, with an average count of 1280. This dataset is diverse in scene distribution and challenging for detection. We use the 5-fold cross-validation method, similar to other papers. The ShanghaiTech Part A dataset

is randomly collected from the internet, containing 300 training images and 182 test images. Part B is captured from busy streets in the Shanghai metropolis, with 400 training images and 316 test images. The crowd density varies significantly between Part A and Part B, making this dataset more challenging and representative than most existing datasets. Additionally, we collected a non-public Tower dataset for practical mid-to-high viewpoint video surveillance applications, consisting of 671 training images and 100 test images.

Similar to previous works, we apply random scaling, keeping the shorter side no less than 128. We then randomly crop 8 fixed-size 128×128 image patches from the resized images. Finally, we introduce random flipping with a probability of 0.5. The Adam algorithm with an initial learning rate of 1e-4 is first used for training, followed by switching to the SGD optimizer for fine-tuning the better model [69]. The backbone network is pretrained on ImageNet, similar to previous works.

For the contrastive cropping sample parameter settings, we focus on the training data batch size and the number of patches per image. We set the batch size to 8 and the patch number to 3. Therefore, for each training process, we obtain 24 samples per batch, with labels indicating whether they contain people or not, which can be used for the contrastive learning process.

## Model evaluation

Our method has been compared with state-of-the-art methods on several publicly available crowd counting datasets, as shown in Table 1. We use Mean Absolute Error (MAE) and Root Mean Squared Error (MSE) as evaluation metrics, defined as follows.

$$MAE = \frac{1}{N} \sum_{i=1}^{N} |y_i - \hat{y}_i| \quad MSE = \sqrt{\frac{1}{N} \sum_{i=1}^{N} (y_i - \hat{y}_i)^2} \tag{10}$$

where $N$ is the number of test images, $y_i$ is the actual number of people in the $i$-th image, and $\hat{y}_i$ is the estimated number of people in the $i$-th image. Roughly speaking, MAE measures the accuracy of the estimates, while MSE measures the robustness of the estimates.

**Table 1. The overall performance of our framework.**

| Methods | SHTech PartA | | SHTech PartB | | UCF_CC_50 | |
|---|---|---|---|---|---|---|
| | MAE | MSE | MAE | MSE | MAE | MSE |
| CAN [71] | 62.3 | 100.0 | 7.8 | 12.2 | 212.2 | **243.7** |
| Bayesian+ [72] | 62.8 | 101.8 | 7.7 | 12.7 | 229.3 | 308.2 |
| S-DCNet [28] | 58.3 | 95.0 | 6.7 | 10.7 | 204.2 | 301.3 |
| SANet+SPANet [73] | 59.4 | 92.5 | 6.5 | 9.9 | 232.6 | 311.7 |
| SDANet [29] | 63.6 | 101.8 | 7.8 | 10.2 | 227.6 | 316.4 |
| ADSCNet [30] | 55.4 | 97.7 | 6.4 | 11.3 | 198.4 | 267.3 |
| AMRNet [31] | 61.59 | 98.36 | 7.02 | 11.0 | 184.0 | 265.8 |
| AMSNet [74] | 56.7 | 93.4 | 6.7 | 10.2 | 208.4 | 297.3 |
| DM-Count [75] | 59.7 | 95.7 | 7.4 | 11.8 | 211.0 | 291.5 |
| P2PNet [40] | 52.74 | 85.06 | 6.25 | 9.9 | **172.7** | 256.2 |
| Chfl [76] | 57.5 | 94.3 | 6.9 | 11.0 | – | – |
| RSI-ResNet50 [70] | 54.8 | 89.1 | 6.2 | **9.9** | 186.3 | 256.5 |
| STEERER-VGG19 [77] | 55.6 | 87.3 | 6.8 | 10.7 | – | – |
| Ours | **51.34** | **83.9** | **6.08** | 10.4 | 173.5 | 248.9 |

Our framework achieves the best performance among all methods. Specifically, on the dense crowd dataset SHTech Part A, our method reduces the MAE by 1.4 and the MSE by 1.16 compared to the state-of-the-art method P2PNet [40]. On the sparse crowd dataset SHTech Part B, our method also achieves the best performance, reducing the MAE by 0.12 compared to the state-of-the-art method RSI-ResNet50 [70]. On the challenging UCF_CC_50 dataset with a wide range of crowd densities, our method balances accuracy and robustness. Compared to P2PNet, our method reduces the MSE by 7.3 while maintaining a similar MAE, and compared to CAN, our method reduces the MAE by 38.7 while maintaining a similar MSE. Considering both MAE and MSE metrics, our method not only achieves high accuracy but also demonstrates high robustness, achieving the best overall performance.

As shown in Fig 4, we present the visualized crowd counting results based on the ShanghaiTech Part A and Part B datasets. In the ShanghaiTech Part A dataset, we observe densely crowded regions, while the ShanghaiTech Part B dataset is more sparsely populated. The images in ShanghaiTech Part A have higher densities, with the number of people ranging from 33 to 3139, whereas the images in ShanghaiTech Part B have lower densities, with the number of people ranging from 9 to 578 [36]. Due to the varying object shapes and sizes in these datasets, our multi-scale feature fusion model demonstrates significant advantages over traditional methods. Experiments on these datasets demonstrate that our method performs well in both sparse and crowded crowd scenarios.

Additionally, we evaluated our model using the Tower dataset. As shown in Fig 5, our method accurately predicts the number of people in both near-field images with fewer people and far-field images with more people. This indicates that our algorithm is adaptable to both near-field and far-field, as well as sparse and dense crowd scenarios.

## Ablation experiments

Considering that the main innovations of the paper revolve around contrastive learning and the multi-scale feature fusion, our ablation experiments focus on four aspects: the effectiveness of the projection head and contrastive loss related to contrastive learning, the effectiveness of the multi-scale feature fusion module, and the number of patches cropped from a single image.

**Effectiveness of the projection head.** Given that different types of projection heads have varying performance, we conducted an ablation study on the structure of the projection head, including three settings: identity mapping, linear projection head using a fully connected (FC) layer of size 256*128, and nonlinear projection head using a 256*256 FC layer, a 256*128 FC layer, and a ReLU activation in between. As shown in Table 2, under the same framework with the patch number set to 4 by default, the linear projection head shows a certain advantage. The learnable linear transformation between the representations and the contrastive loss significantly improves the quality of learned representations.

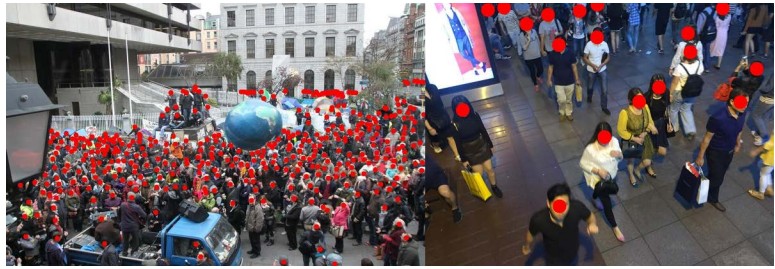

**Fig 4. Visualization of our method on the ShanghaiTech Part A and Part B datasets.** The left image is from ShanghaiTech Part A, with a predicted crowd count of 375; the right image is from Part B, with a predicted crowd count of 18. All faces are blurred in Fig 4 for privacy preservation.

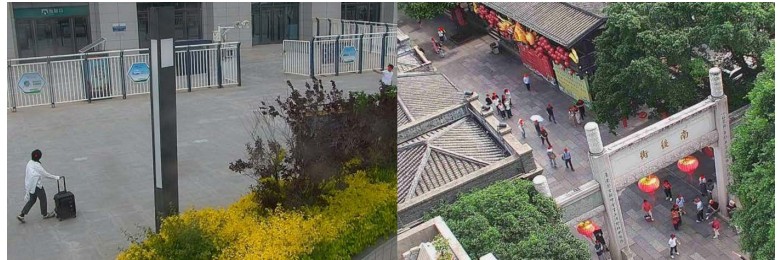

**Fig 5. Example visualization results of our method on the Tower dataset.** The left image shows a close-up view of a high-speed rail station exit, with a predicted crowd count of 2; the right image shows a distant view of a street, with a predicted crowd count of 33. All faces are blurred in Fig 5 for privacy preservation.

**Table 2. Ablation study on projection head.**

| SHTechA dataset | MAE | MSE |
|---|---|---|
| Identity mapping | 52.05 | 83.8 |
| Non-linear projection | 51.86 | 85.09 |
| Linear projection | **51.84** | **83.39** |

**Table 3. Evaluation of the effectiveness of MSFM.**

| SHTechA dataset | MAE | MSE |
|---|---|---|
| Backbone+FPN | 53.54 | 87 |
| Backbone+MSFM | **52.36** | **84.55** |

**Effectiveness of the multi-scale feature fusion module.** To analyze the impact of feature fusion, we compared the original FPN module with our Multi-Scale Feature Fusion Module (MSFM). Using the same VGG backbone network and the default hyperparameter settings mentioned above, the experimental results are shown in Table 3. The multi-scale feature fusion module reduces the MAE by 1.18 and the MSE by 2.45 compared to the original FPN module. As we can see, by replacing FPN with our proposed MSFM, we achieved better crowd counting performance. The results indicate that multi-scale feature fusion can extract scale-related features from crowd images, facilitating crowd detection in different contexts.

**Effectiveness of the contrastive loss.** We conducted an ablation study on the use of contrastive loss, and the analysis results are shown in Table 4. To ensure the reliability of the experiments, the baseline model was based on the same model without contrastive loss, with the patch number set to 4 by default. We can see that adding contrastive loss reduces the MAE by 0.52 and the MSE by 1.16, showing that the model with contrastive loss performs better.

**Patch number parameter analysis.** Under uniform parameter configurations, experiments were conducted on the SHTechA dataset to investigate the influence of the contrastive cropping module on crowd counting by varying the number of patches cropped from a single image. As depicted in Fig 6, when the patch number is set to 3, the crowd counting accuracy, measured by mean absolute error (MAE), attains an optimal value of 51.34. This outcome is attributable to the ability of a moderate patch count to enable the contrastive learning framework to effectively capture multi-scale contextual information while maintaining a balanced distribution of positive/negative sample pairs, which is crucial for learning discriminative features in crowd scenes. As the patch number exceeds 3, significant fluctuations in the MAE metric emerge, primarily due to the introduction of redundant or semantically similar regions from excessive patches,

**Table 4. Evaluation of the effectiveness of contrastive loss.**

| SHTechA dataset | MAE | MSE |
|---|---|---|
| Baseline | 52.36 | 84.55 |
| Baseline+Conloss | **51.84** | **83.39** |

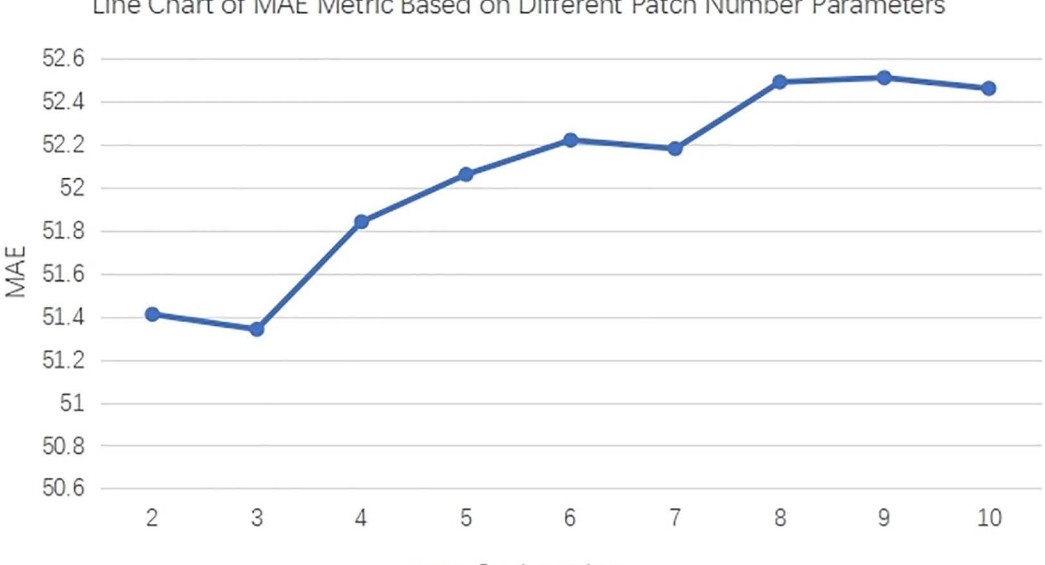

**Fig 6. Evaluation of MAE accuracy metric based on different patch numbers for our method on the SHTechA crowd counting dataset.** Patch number parameter is the number of samples cropped from a single image for contrastive learning.

which dilutes the informativeness of contrastive pairs. In contrastive learning, the efficacy of the loss function hinges on meaningful semantic disparities between sample pairs; thus, an overly high patch number forces the model to learn from highly correlated patches, leading to unstable gradient updates and degraded feature representation quality. Specifically, when the patch number surpasses 8, the overload of intra-image patches disrupts the equilibrium of the contrastive learning objective, causing the MAE to exceed the baseline value of 52.36. These findings underscore the necessity of constraining the patch number within a range below 8 to optimize contrastive feature diversity while mitigating information redundancy, thereby validating the theoretical rationale of the proposed contrastive cropping module. By adaptively selecting an appropriate number of patches, the module balances multi-scale feature extraction with contrastive sample quality, enhancing the discriminative power of learned representations and consequently improving crowd counting performance.

## Qualitative study on contrastive learning

To obtain more insights into the effectiveness of contrastive learning, we apply t-SNE [78] to MSFM features by different models to compare their capability to distinguish between positive samples and negative samples. Fig 7 presents t-SNE maps for the baseline model and our model based on contrastive learning. The t-SNE map of features after the MSFM module integrated with contrastive learning shows more separability than the baseline t-SNE map.

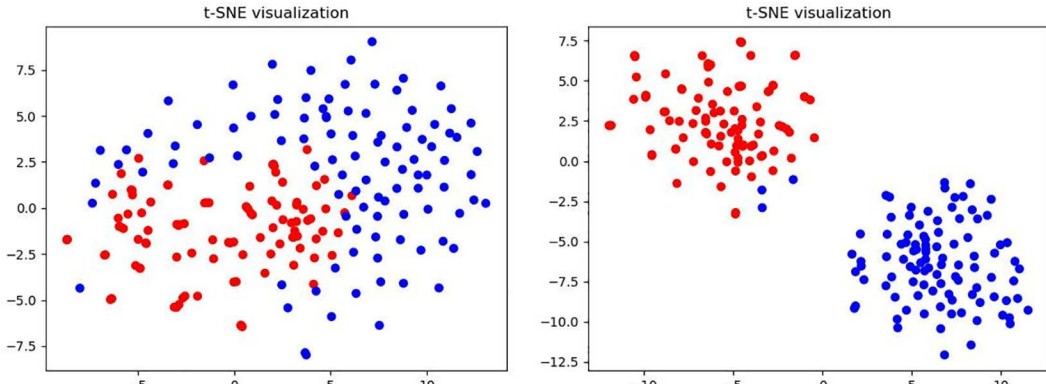

**Fig 7. The t-SNE map of MSFM features by different models, in which the blue dots and red dots refer to the positive samples and negative samples.** Left: features after the MSFM model integrated with contrastive learning; Right: features of the baseline model.

**Table 5. Comparison of the Parameters (M) and Inference speed (s/100 images).**

| Model | Parameters | Inference speed |
|---|---|---|
| STEERER-VGG19 [77] | 64.64 | 2.54 |
| Ours | **21.58** | **0.38** |

## Complexity analysis

Table 5 reports a comparison of model size and inference speed computed with a single NVIDIA T4 GPU. The inference time is the average time of 100 runs on testing 1920×1080 sample. Those test images are collected from real-world monitoring scenes. It can be observed that our method has less parameters and faster inference speed than the previous state-of-the-art method STEERER [77]. Evidently, our model excels in addressing real-world scenarios in terms of both resource consumption and inference performance.

## Conclusion

This paper proposes a novel and effective point-based crowd counting method using contrastive learning. Recent studies have shown that detecting dense small objects is a significant and challenging problem in public safety and video surveillance. To address this issue, we introduce a point-based crowd detection framework and leverage auxiliary supervised contrastive learning to enhance the model's ability to represent crowd foreground and background. Additionally, a multi-scale feature fusion module is proposed for detecting crowds in both high-density and low-density regions. Several experiments conducted on public crowd datasets demonstrate the effectiveness of our method. To further improve the detection accuracy in dense crowd scenes, other deep learning models such as Transformers could be considered to enhance the feature extraction capability. In the future, our method can also be applied to count other objects, such as vehicles and animals, and analyze their behaviors based on the point locations obtained by our model.

## Author contributions

**Conceptualization:** Rundong Cao, Jiazhong Yu.

**Data curation:** Rundong Cao.

**Formal analysis:** Rundong Cao.

**Funding acquisition:** Jiazhong Yu.

**Investigation:** Rundong Cao.

**Methodology:** Rundong Cao.

**Project administration:** Rundong Cao, Jiazhong Yu.

**Resources:** Rundong Cao, Jiazhong Yu.

**Software:** Rundong Cao.

**Supervision:** Jiazhong Yu, Ziwei Liu, Qinghua Liang.

**Validation:** Rundong Cao.

**Visualization:** Rundong Cao.

**Writing – original draft:** Rundong Cao.

**Writing – review & editing:** Rundong Cao, Ziwei Liu, Qinghua Liang.

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
