## [Decision Letter · Decision Letter 0]

Dear Dr. Cao,

Thank you for submitting your manuscript to PLOS ONE. After careful consideration, we feel that it has merit but does not fully meet PLOS ONE’s publication criteria as it currently stands. Therefore, we invite you to submit a revised version of the manuscript that addresses the points raised during the review process.

We look forward to receiving your revised manuscript.

Kind regards,

Yawen Lu, Ph.D

Academic Editor

PLOS ONE

Additional Editor Comments :

Dear Dr. Cao,

PONE-D-24-51986

I am writing to you regarding the above referenced manuscript that you submitted to Plos One.

Based on the enclosed reviews, I am pleased to inform you that this manuscript is recommended for Major Revision for publication in the journal.

Please carefully address the reviewers' comments and suggestions regarding network efficiency, used data, organization, explanation, and literature reviews, etc., to improve the quality of the manuscript in the revised submission.

Reviewers' comments:

Reviewer's Responses to Questions

**Comments to the Author**

1. Is the manuscript technically sound, and do the data support the conclusions?

Reviewer #1: Partly

Reviewer #2: Partly

2. Has the statistical analysis been performed appropriately and rigorously?

Reviewer #1: Yes

Reviewer #2: No

3. Have the authors made all data underlying the findings in their manuscript fully available?

Reviewer #1: Yes

Reviewer #2: Yes

4. Is the manuscript presented in an intelligible fashion and written in standard English?

Reviewer #1: Yes

Reviewer #2: No

Reviewer #1: 1. The point prediction method mentioned in the paper has enhanced detection in both high- and low-density regions, but adaptability in complex scenarios is still a concern. Is the introduction of more training data or the use of data augmentation techniques considered to improve the robustness of the model? For example, training the model with different lighting conditions, weather changes, or different viewpoints may make it perform more robustly in real-world applications.

2. How to further improve the detection accuracy in dense crowd scenes is an important research direction. The multi-scale feature fusion module mentioned in the paper is already a good attempt, but is it possible to explore other detection methods or optimize the existing framework? For example, it is possible to consider combining other deep learning models (e.g., the Transformer architecture) to enhance the feature extraction capability, or to introduce self-supervised learning methods to further improve the performance of the model.

3. The model mentioned in the paper directly predicts header points during inference, which may affect real-time performance. Is it possible to optimize the computational efficiency of the model for real-time monitoring while maintaining accuracy?

4. It is mentioned in the paper that the method can be applied to the counting of other objects, such as vehicles and animals, in the future. Are there any plans to conduct relevant experiments to verify the effectiveness of the method in counting different objects? In addition, is it considered to combine the method with behavioral analysis to provide a more comprehensive monitoring solution?

Reviewer #2: This paper presents a novel approach to crowd counting using contrastive learning and point-based detection. While the work shows promise and addresses important challenges in crowd counting, several major issues need to be addressed.

Strengths:

- The paper addresses important real-world challenges in crowd counting - dealing with complex backgrounds and multi-scale targets.

- The authors provide comprehensive experiments across several datasets to validate the proposed method's effectiveness.

- The experimental results show the approach's state-of-the-art performance.

Suggestions:

- The current experiments and analysis do not sufficiently demonstrate the contribution of contrastive learning. Please provide more thorough validation and ablation studies.

- The paper organization needs significant improvement. Important technical details are scattered and sometimes unclear.

- Since the paper focuses on a real-world problem, the computational complexity analysis and comparison should be considered.

- Please provide more detailed explanations and annotations for the figures (e.g., figure 1, 3, 6)

- Current literature reviews missed some recent works on local-global feature fusion and point/feature matching in visual learning, including [Transflow: Transformer as flow learner, 2023][Fusioncount: Efficient crowd counting via multiscale feature fusion, 2022][Lightglue: Local feature matching at light speed, 2023][Superthermal: Matching thermal as visible through thermal feature exploration, 2021], etc.

**Do you want your identity to be public for this peer review?** For information about this choice, including consent withdrawal, please see our Privacy Policy

Reviewer #1: No

Reviewer #2: No

---

## [Author Response · Author response to Decision Letter 1]

19 Feb 2025

Dear Editors and Reviewers:

Thank you for your letter and for the reviewers’ comments concerning our manuscript entitled “Towards Real-world Monitoring Scenarios: An Improved Point Prediction Method for Crowd Counting Based on Contrastive Learning” (No.: PONE-D-24-51986). Those comments are all valuable and very helpful for revising and improving our paper, as well as the important guiding significance to our researches. We have studied comments carefully and have made correction which we hope meet with approval.

Reviewers' comments:

Reviewer's Responses to Questions

1. Is the manuscript technically sound, and do the data support the conclusions?

Reviewer #1: Partly

Reviewer #2: Partly

The authors’ answer:

We sincerely appreciate the reviewers for their acknowledgment of our paper. We have made every effort to improve the manuscript and have implemented several changes.

2. Has the statistical analysis been performed appropriately and rigorously?

Reviewer #1: Yes

Reviewer #2: No

The authors’ answer:

We sincerely appreciate the reviewers for their recognition of our paper. We have made every effort to improve the manuscript and have implemented several changes.

3. Have the authors made all data underlying the findings in their manuscript fully available?

Reviewer #1: Yes

Reviewer #2: Yes

The authors’ answer:

We sincerely appreciate the reviewers for acknowledging our work.

4.Is the manuscript presented in an intelligible fashion and written in standard English?

Reviewer #1: Yes

Reviewer #2: No

The authors’ answer:

Thanks for your suggestion. We have made every effort to improve the manuscript and have implemented several changes. These modifications will not affect the content or structure of the paper.

5. Review Comments to the Author

Reviewer #1:

1.The point prediction method mentioned in the paper has enhanced detection in both high- and low-density regions, but adaptability in complex scenarios is still a concern. Is the introduction of more training data or the use of data augmentation techniques considered to improve the robustness of the model? For example, training the model with different lighting conditions, weather changes, or different viewpoints may make it perform more robustly in real-world applications.

The authors’ answer:

We have used more training data and data augmentation techniques in practical applications, but this part was not included in the paper to maintain consistency with other methods. Considering the reviewer's suggestion, we have added this part, which is highlighted in red in the Introduction section.

2.How to further improve the detection accuracy in dense crowd scenes is an important research direction. The multi-scale feature fusion module mentioned in the paper is already a good attempt, but is it possible to explore other detection methods or optimize the existing framework? For example, it is possible to consider combining other deep learning models (e.g., the Transformer architecture) to enhance the feature extraction capability, or to introduce self-supervised learning methods to further improve the performance of the model.

The authors’ answer:

It is possible to explore other detection methods or optimize the existing framework. Considering the reviewer's suggestion, we have added a discussion on this issue , which is highlighted in red in the Conclusion section.

3. The model mentioned in the paper directly predicts header points during inference, which may affect real-time performance. Is it possible to optimize the computational efficiency of the model for real-time monitoring while maintaining accuracy?

The authors’ answer:

The model we designed in the paper has already considered computational efficiency of the model for real-time monitoring. Our network is faster than other deep learning models such as Transformers. Besides, our method consumes less GPU memory, which makes practical applications more advantageous.

4.It is mentioned in the paper that the method can be applied to the counting of other objects, such as vehicles and animals, in the future. Are there any plans to conduct relevant experiments to verify the effectiveness of the method in counting different objects? In addition, is it considered to combine the method with behavioral analysis to provide a more comprehensive monitoring solution?

The authors’ answer:

Thanks for your suggestion. We have plans to conduct relevant experiments for counting different objects and combine the method with behavioral analysis in the future. However, we will not do the experiments in this paper due to the limitations of research conditions and article length.

Reviewer #2: This paper presents a novel approach to crowd counting using contrastive learning and point-based detection. While the work shows promise and addresses important challenges in crowd counting, several major issues need to be addressed.

- The current experiments and analysis do not sufficiently demonstrate the contribution of contrastive learning. Please provide more thorough validation and ablation studies.

The authors’ answer:

Thank you for your suggestion. We have already conducted an ablation study on the use of contrastive loss to demonstrate the contribution of contrastive learning. And we provide more thorough validation and ablation studies which is highlighted in red in the Qualitative Study on Contrastive learning part.

- The paper organization needs significant improvement. Important technical details are scattered and sometimes unclear.

The authors’ answer:

Thanks for your suggestion. We have revised the paper organization highlighted in red in the Our Method section.

- Since the paper focuses on a real-world problem, the computational complexity analysis and comparison should be considered.

The authors’ answer:

Thanks for your suggestion. We have added the computational complexity analysis and compared with other methods. This addition is marked in red in the Experiments section.

- Please provide more detailed explanations and annotations for the figures (e.g., figure 1, 3, 6)

The authors’ answer:

We have already provided explanations and annotations for the figures. Considering the reviewer's suggestion, we have added more explanations highlighted in red for the figures including figure 1, 3, 6.

- Current literature reviews missed some recent works on local-global feature fusion and point/feature matching in visual learning, including [Transflow: Transformer as flow learner, 2023][Fusioncount: Efficient crowd counting via multiscale feature fusion, 2022][Lightglue: Local feature matching at light speed, 2023][Superthermal: Matching thermal as visible through thermal feature exploration, 2021], etc.

The authors’ answer:

Thanks for your suggestion. We have revised the literature reviews highlighted in red in the Introduction and Related Work section.

---

## [Decision Letter · Decision Letter 1]

Dear Dr. Cao,

Thank you for submitting your manuscript to PLOS ONE. After careful consideration, we feel that it has merit but does not fully meet PLOS ONE’s publication criteria as it currently stands. Therefore, we invite you to submit a revised version of the manuscript that addresses the points raised during the review process.

Your manuscript presents a well-structured analysis, but a few areas require minor revisions for improved clarity and consistency.

We look forward to receiving your revised manuscript.

Kind regards,

Ayesha Maqbool, PhD

Academic Editor

PLOS ONE

Journal Requirements:

Reviewers' comments:

Reviewer's Responses to Questions

**Comments to the Author**

Reviewer #2: (No Response)

Reviewer #3: All comments have been addressed

2. Is the manuscript technically sound, and do the data support the conclusions?

Reviewer #2: Partly

Reviewer #3: (No Response)

3. Has the statistical analysis been performed appropriately and rigorously?

Reviewer #2: No

Reviewer #3: (No Response)

4. Have the authors made all data underlying the findings in their manuscript fully available?

Reviewer #2: Yes

Reviewer #3: (No Response)

5. Is the manuscript presented in an intelligible fashion and written in standard English?

Reviewer #2: No

Reviewer #3: (No Response)

Reviewer #2: This paper presents a novel approach to crowd counting using contrastive learning and point-based detection. The topic is interesting and meaningful in real-world. However, some of my concerns have not been addressed.

1. Some captions for the figures are too simple, the authors should provide an indepth analysis for each figure. For example, while the Fig. 6 illustrates the impact of different patch numbers on MAE, the caption should further explain key trends, such as why the MAE fluctuates after a certain threshold and how this finding supports the proposed method.

2. In Figure 1, the point-based detection results are really hard to recognize. The authors may consider changing the point color.

3. Some sections could be better structured for clarity, particularly in the explanation of contrastive learning.

4. I recommend a thorough revision of the writing to improve readability.

Reviewer #3: Thank you for taking the time to consider and thoughtfully address my comments. I appreciate your attention to detail and your commitment to incorporating constructive feedback to improve the overall quality of the work

**Do you want your identity to be public for this peer review?** For information about this choice, including consent withdrawal, please see our Privacy Policy

Reviewer #2: No

Reviewer #3: No

---

## [Author Response · Author response to Decision Letter 2]

11 Jun 2025

Dear Editors and Reviewers:

Thank you for your letter and for the reviewers’ comments concerning our manuscript entitled “Towards Real-world Monitoring Scenarios: An Improved Point Prediction Method for Crowd Counting Based on Contrastive Learning” (No.: PONE-D-24-51986R1). Those comments are all valuable and very helpful for revising and improving our paper, as well as the important guiding significance to our researches. We have carefully studied the comments and made the necessary corrections, hoping that they will meet with your approval.

Reviewers' comments:

Reviewer's Responses to Questions

1. If the authors have adequately addressed your comments raised in a previous round of review and you feel that this manuscript is now acceptable for publication, you may indicate that here to bypass the “Comments to the Author” section, enter your conflict of interest statement in the “Confidential to Editor” section, and submit your "Accept" recommendation.

Reviewer #2: (No Response)

Reviewer #3: All comments have been addressed

The authors’ answer:

We sincerely appreciate the reviewers for their acknowledgment of our paper.

2. Is the manuscript technically sound, and do the data support the conclusions?

Reviewer #2: Partly

Reviewer #3:

The authors’ answer:

We sincerely appreciate the reviewers for their acknowledgment of our paper. We have made every effort to improve the manuscript and have implemented several changes.

3. Has the statistical analysis been performed appropriately and rigorously?

Reviewer #2: No

Reviewer #3:

The authors’ answer:

We sincerely appreciate the reviewers for their recognition of our paper. We have made every effort to improve the manuscript and have implemented several changes.

4. Have the authors made all data underlying the findings in their manuscript fully available?

Reviewer #2: Yes

Reviewer #3:

The authors’ answer:

We sincerely appreciate the reviewers for acknowledging our work.

5. Is the manuscript presented in an intelligible fashion and written in standard English?

Reviewer #2: No

Reviewer #3:

The authors’ answer:

Thanks for your suggestion. We have made every effort to improve the manuscript and have implemented several changes. These modifications will not affect the content or structure of the paper.

6. Review Comments to the Author

Reviewer #2:

1. Some captions for the figures are too simple, the authors should provide an indepth analysis for each figure. For example, while the Fig. 6 illustrates the impact of different patch numbers on MAE, the caption should further explain key trends, such as why the MAE fluctuates after a certain threshold and how this finding supports the proposed method.

The authors’ answer:

Thanks for your suggestion. We have revised the illustration, highlighted in red, for explaining key trends in MAE of different patch numbers in the Ablation Experiments section.

2. In Figure 1, the point-based detection results are really hard to recognize. The authors may consider changing the point color.

The authors’ answer:

Thanks for your suggestion. We have changed the point color and point size to make the point-based results easier to recognize.

3. Some sections could be better structured for clarity, particularly in the explanation of contrastive learning.

The authors’ answer:

Thanks for your suggestion. We have modified the paper structure highlighted in red in the Our Method section. In the section on contrastive learning network design and loss function design, we elucidate the methodology of leveraging contrastive learning to enhance the performance of crowd counting.

4. I recommend a thorough revision of the writing to improve readability.

The authors’ answer:

Thanks for your suggestion. We have made every effort to improve the manuscript and hope to meet with approval.

Reviewer #3: Thank you for taking the time to consider and thoughtfully address my comments. I appreciate your attention to detail and your commitment to incorporating constructive feedback to improve the overall quality of the work

The authors’ answer:

We sincerely appreciate your acknowledgment of our paper.

---

## [Editor Report · Decision Letter 2]

Towards Real-world Monitoring Scenarios: An Improved Point Prediction Method for Crowd Counting Based on Contrastive Learning

PONE-D-24-51986R2

Dear Dr. Cao,

We’re pleased to inform you that your manuscript has been judged scientifically suitable for publication and will be formally accepted for publication once it meets all outstanding technical requirements.

Kind regards,

Ayesha Maqbool, PhD

Academic Editor

PLOS ONE
---

## [Editor Report · Acceptance letter]

PONE-D-24-51986R2

PLOS ONE

Dear Dr. Cao,

I'm pleased to inform you that your manuscript has been deemed suitable for publication in PLOS ONE. Congratulations! Your manuscript is now being handed over to our production team.

Kind regards,

on behalf of

Dr. Ayesha Maqbool

Academic Editor

PLOS ONE